# Effects of *Allium hookeri* on gut microbiome related to growth performance in young broiler chickens

Sung-Hyen Lee[1]*, Sohyun Bang[2], Hwan-Hee Jang[1], Eun-Byeol Lee[1], Bong-Sang Kim[3], Seung-Hwan Kim[4], Sang-Hyun Kang[4], Kyung-Woo Lee[5], Dong-Wook Kim[6], Jung-Bong Kim[1], Jeong-Sook Choe[1], Shin-Young Park[1], Hyun S. Lillehoj[7]

1 National Institute of Agricultural Sciences, Rural Development Administration, Isoe-myeon, Wanju-Gun, Jeollabuk-do, Republic of Korea, 2 Interdisciplinary Program in Bioinformatics, Seoul National University, Kwan-ak Gu, Seoul, Republic of Korea, 3 Department of Agricultural and Life Sciences and Research Institute of Population Genomics, Seoul National University, Seoul, Republic of Korea, 4 KYOCHON F&B CO, Osan city, Kyounggido, Republic of Korea, 5 Department of Animal Science and Technology, Konkuk University, Gawngjin-gu, Seoul, Republic of Korea, 6 Department of Poultry Science, Korean National College of Agriculture and Fisheries, Deokjin-gu, Jeonju-si, Jeollabuk-do, Republic of Korea, 7 Animal Bioscience and Biotechnology Laboratory, Beltsville Agricultural Research Center, Agricultural Research Service, Department of Agriculture, Beltsville, MD, United States of America

☯ These authors contributed equally to this work.
* sunghyeonl@yahoo.co.kr

**Data Availability Statement:** All relevant data are within the manuscript and its Supporting Information files.

## Abstract

Healthy food promotes beneficial bacteria in the gut microbiome. A few prebiotics act as food supplements to increase fermentation by beneficial bacteria, which enhance the host immune system and health. *Allium hookeri* is a healthy food with antioxidant and anti-inflammatory activities. *A. hookeri* is used as a feed supplement for broiler chickens to improve growth performance. Although the underlying mechanism is unknown, *A. hookeri* may alter the gut microbiome. In the current study, 16S rRNA sequencing has been carried out using samples obtained from the cecum of broiler chickens exposed to diets comprising different tissue types (leaf and root) and varying amounts (0.3% and 0.5%) of *A. hookeri* to investigate their impact on gut microbiome. The microbiome composition in the groups supplemented with *A. hookeri* leaf varied from that of the control group. Especially, exposure to 0.5% amounts of leaf resulted in differences in the abundance of genera compared with diets comprising 0.3% leaf. Exposure to a diet containing 0.5% *A. hookeri* leaf decreased the abundance of the following bacteria: *Eubacterium nodatum*, *Marvinbryantia*, *Oscillospira*, and *Gelria*. The modulation of gut microbiome by leaf supplement correlated with growth traits including body weight, bone strength, and infectious bursal disease antibody. The results demonstrate that *A. hookeri* may improve the health benefits of broiler chickens by altering the gut microbiome.

## Introduction

Diet plays an important role in modulating gut microbiome by providing food substrates for gut microorganisms [1,2]. Dietary components not digestible by host enzymes can be digested

**Funding:** This research was fully supported by the fund of Agriculture Science & Technology Development (PJ01178705, PJ01327901) from National Institute of Agricultural Sciences, Rural Development Administration, Republic of Korea. The fund was used in this study design, data collection and analysis, or preparation of the manuscript. Bong-Sang Kim received support in the form of a salary from C&K genomics. Seung-Hwan Kim and Sang-Hyun Kang received support in the form of salaries from BHNBIO (KYOCHON F&B CO). The specific roles of the authors are articulated in the 'author contributions' section.

**Competing interests:** The authors have read the journal's policy and have the following conflicts: Bong-Sang Kim was employed by C&K genomics at the time of the study. Seung-Hwan Kim and Sang-Hyun Kang are employed by BHNBIO (KYOCHON F&B CO). This does not alter our adherence to all the PLOS ONE policies on sharing data and materials.

by gut bacteria [3]. For example, prebiotics such as inulin, polyphenol, and galacto-oligosaccharide are non-digestible food ingredients that promote the growth of beneficial bacteria [4]. Such prebiotics increase the fermentation products produced by beneficial bacteria, which enhances host immune response [5,6]. Elucidation of the interactions between diet and microbiome has raised the interest in functional foods with beneficial effects on gut microbiome and host health [7].

Among the various functional foods, *Allium hookeri* is widely used as a health food that treat high blood glucose or lipid levels in patients with diabetes mellitus in Korea [8,9]. *A. hookeri* belongs to the genus *Allium*, which includes *A. cepa* (onion) and *A. sativum L.* (garlic). *Allium* species have been used as medicinal foods to reduce the risk of several types of cancers by preventing mutagenesis [10]. The beneficial effects of *Allium* are attributed to the abundance of organosulfur compounds, polyphenols, and allicin [11,12]. *A. hookeri* contains six-fold higher levels of organosulfur than garlic, and higher cellulose and total phenol contents than onion [13]. As these components exhibit antioxidant activities, the use of *A. hookeri* is promising as a medical food [14,15]. *A. hookeri* exhibits immunomodulatory effects in lymphocytes, macrophages, and tumor cells in *in-vitro* chicken cell experiments [16]. *In vivo* experiments have also suggested that *A. hookeri* inhibits the inflammatory response in the pancreas of diabetic rats and LPS-induced young broiler chickens [17,18].

The beneficial effects of *A. hookeri* on health suggest its use in commercial animal farming including pigs and chickens [19–21]. *A. hookeri*, when used as a feed supplement, enhanced the oxidative stability of pork and improved the growth performance of broiler chickens [22]. Although the mechanism of action is unclear it is suspected that *A. hookeri* alters the gut microbiome. *A. hookeri* components such as organosulfur compounds, polyphenols, and allicin are known to affect the gut microbiome by increasing or decreasing the bacterial composition [21,23,24]. Further, a previous study has reported that diets including onions belonging to *Allium* genus modulate gut microbiota and increase body weights of broiler chickens [25,26]. Thus, it has been hypothesized that *A. hookeri* alters the gut microbiome and that such changes might lead to beneficial growth effects in commercial animals. However, the collective effect of *A. hookeri* on gut microorganisms needs to be further elucidated.

The objective of this study was to determine the effect of *A. hookeri* on gut microbiome in chicken. We investigated the changes in the composition of gut microorganisms in chicken by feeding leaves and roots of *A. hookeri*. We sequenced 24 caecal samples derived from six groups of chickens (four samples each). The groups included chickens fed with individual diets containing 0.3% leaves, 0.5% leaves, 0.3% roots, and 0.5% roots. The effects of *A.hookeri* as a feed additive were evaluated by comparing the groups exposed to *A. hookeri* diet with the control group (Control) or commercial supplement (CS). We examined whether *A. hookeri* altered the composition of microorganisms, and determined their effect on growth performance in chickens.

## Results

### Effects of *A. hookeri* as a feed supplement on gut microbiome diversity

To elucidate the differences in microbiota exposed to different amounts of *A. hookeri* roots or leaves, a principal coordinate analysis (PCoA) based on weighted UniFrac metrics was performed (Fig 1). Samples within the leaf group were clustered at shorter distances compared with those in the other groups. Permutational Multivariate Analysis of Variance (PERMANOVA) was also used to determine the significant differences between groups. The variation between the six groups was observed (P-value < 0.05; Table 1). We also examined the effect of feeding with *A. hookeri* leaf and root on the composition of gut microorganisms by grouping

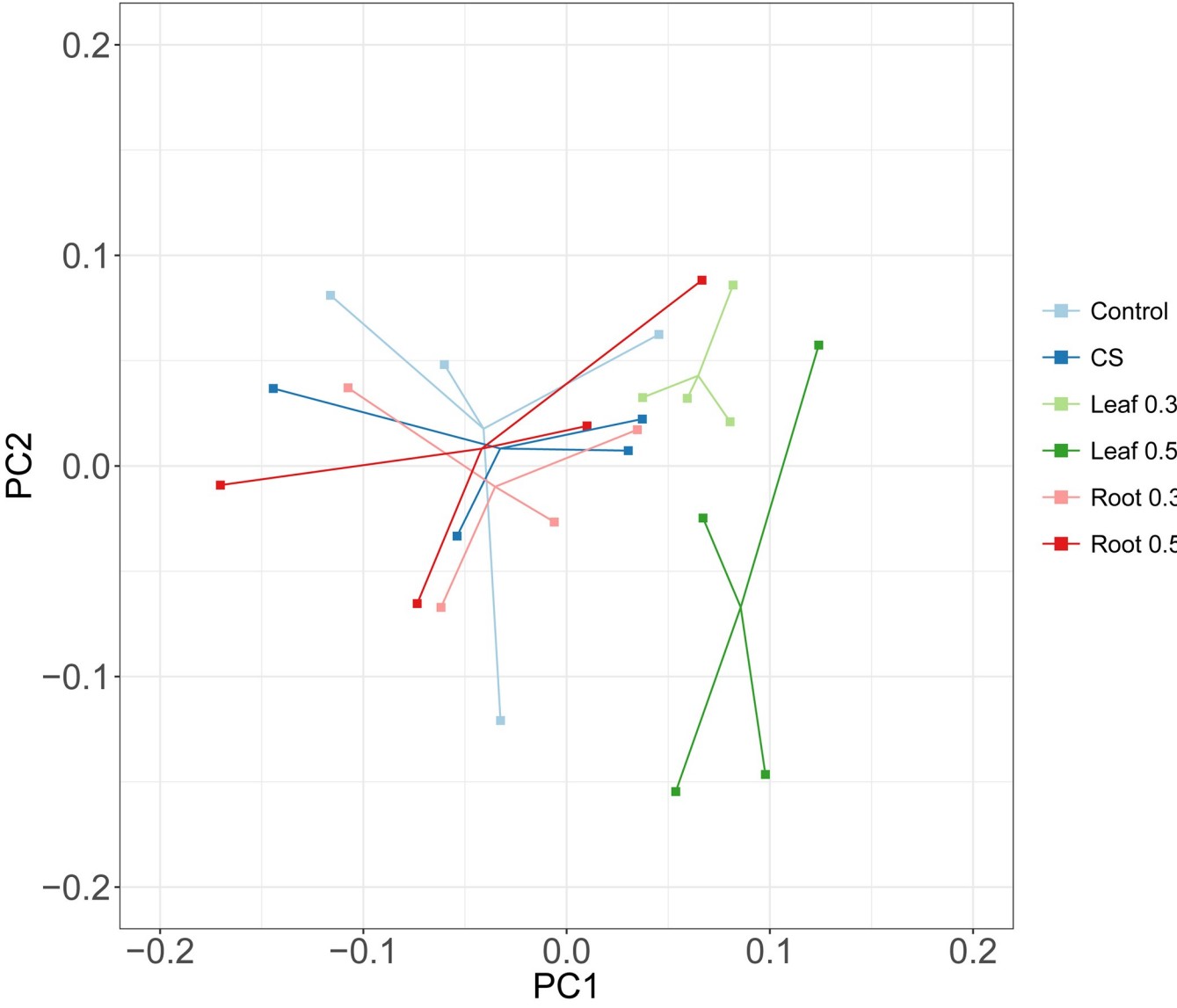

**Fig 1. A principal coordinate analysis (PCoA) plot showing dissimilarities among different diet groups.** PCoA from distance of weighted UniFrac. Each dot represents a single sample. "Leaf", "Root", "0.3", "0.5" denote leaf, root of *A. hookeri*, 0.3%, and 0.5% diets, respectively.

the samples exposed to *A. hookeri* according to the tissue or amount. When we divided the samples exposed to *A. hookeri* leaves and roots irrespective of the amounts of *A. hookeri*, significant differences occurred between Control, CS, Leaf and Root. However, we did not find significant differences between Control, CS, 0.3% and 0.5%, when the samples were divided according to the amounts of *A. hookeri*.

**Table 1. Results of PERMANOVA.**

| Test group | $r^2$ | F-Ratio | P-value |
|---|---|---|---|
| Tissue and amount of *A. hookeri* | 0.27 | 1.34 | 0.016 |
| Tissue of *A. hookeri* | 0.17 | 1.44 | 0.011 |
| Amount of *A. hookeri* | 0.13 | 1.07 | 0.31 |

The groups exhibiting significant variation were subjected to PERMANOVA pair-wise tests as follows: (1) groups of 'Tissue of *A. hookeri*', and (2) 'Tissue and amount of *A. hookeri*'. In the pair-wise test for (1), 'Control, Root' showed no significiant difference, although the difference between control groups and leaf was significant, with a low p-value (Table 2). In the pair-wise test for (2), pairs of groups exposed to the same tissues but different amounts (Leaf 0.3, Leaf 0.5 pair and Root 0.3, Root 0.5 pair) showed no significant difference (S1 Table), which suggested that the effect of 0.3% and 0.5% *A. hookeri* showed little difference as in-feed supplements. Combinations with Leaf 0.5 were significant with lower p-values than the other pair-wise combinations indicating that the abundance and variety of microorganisms in chickens exposed to leaf 0.5 differed from that of the control group.

The diversity of microorganisms within a local community was evaluated based on diversity and richness using Shannon index and the observed operational taxonomic unit (OTU), respectively (S1 Fig). The diversity showed a similar distribution of Shannon index across combinations except for Leaf 0.3 and Root 0.3 compared with CS ($P = 0.02$). Richness also showed similar distribution across groups with 'combinations of tissue and amount' of *A. hookeri* ($P = 0.083$) except for Leaf 0.3 compared with CS ($P = 0.02$). Compared with the control, diversity and richness were not affected by diets supplemented with *A. hookeri* ($P = 0.5$). In summary, while the differences between chickens exposed to leaf and other groups showed better diversity, the differences within a local community were not apparent when compared with the control.

## *A. hookeri*-associated microbiota changes

At the phylum level, *Firmicutes* and *Bacteroidetes* were the major groups of microorganisms in the cecum of chicken (Fig 2A), accounting for more than 80% and 10% of microorganisms, respectively. The amounts of most phyla including *Firmicutes* and *Bacteoidetes* showed no difference between supplements and Control or CS groups except for the composition of *Cyanobacteria* identified as *Gastranaerophilales* at the class level, which was decreased in Leaf 0.5 compared with that of the Control or CS. However, only limited information was available regarding the differences at the phylum level. Therefore, the differences in the abundance of microorganisms at the genus level were investigated.

Approximately 70% of the features exist at the genus level. The dominant genera in the samples are shown in S2 Fig. These features were associated with the differential abundance of microorganisms by comparing each *A. hookeri* supplement group with the Control or CS (FDR< 0.05). The chickens fed with the root supplement did not show differential abundance of taxa. However, a few genera were significantly associated with the gut microbiome of chickens exposed to Leaf compared with the Control or CS categories. Chickens fed with Leaf 0.5 showed seven different genera compared with the Control. Chickens exposed to Leaf 0.5 carried five genera compared with the CS. By contrast, chicken exposed to Leaf 0.3 showed only two different genera compared with the CS. Based on these findings, we concluded that only the Leaf supplement was associated with the abundance of microbial genera. The amounts of leaf supplement also altered the abundance of microorganisms.

**Table 2. Results of PERMANOVA of pair-wise test for tissue groups.**

| Pair | P-value | Pair | P-value |
|---|---|---|---|
| Control, CS | 0.537 | CS, Leaf | 0.018 |
| Control, Leaf | 0.022 | CS, Root | 0.438 |
| Control, Root | 0.056 | Leaf, Root | 0.005 |

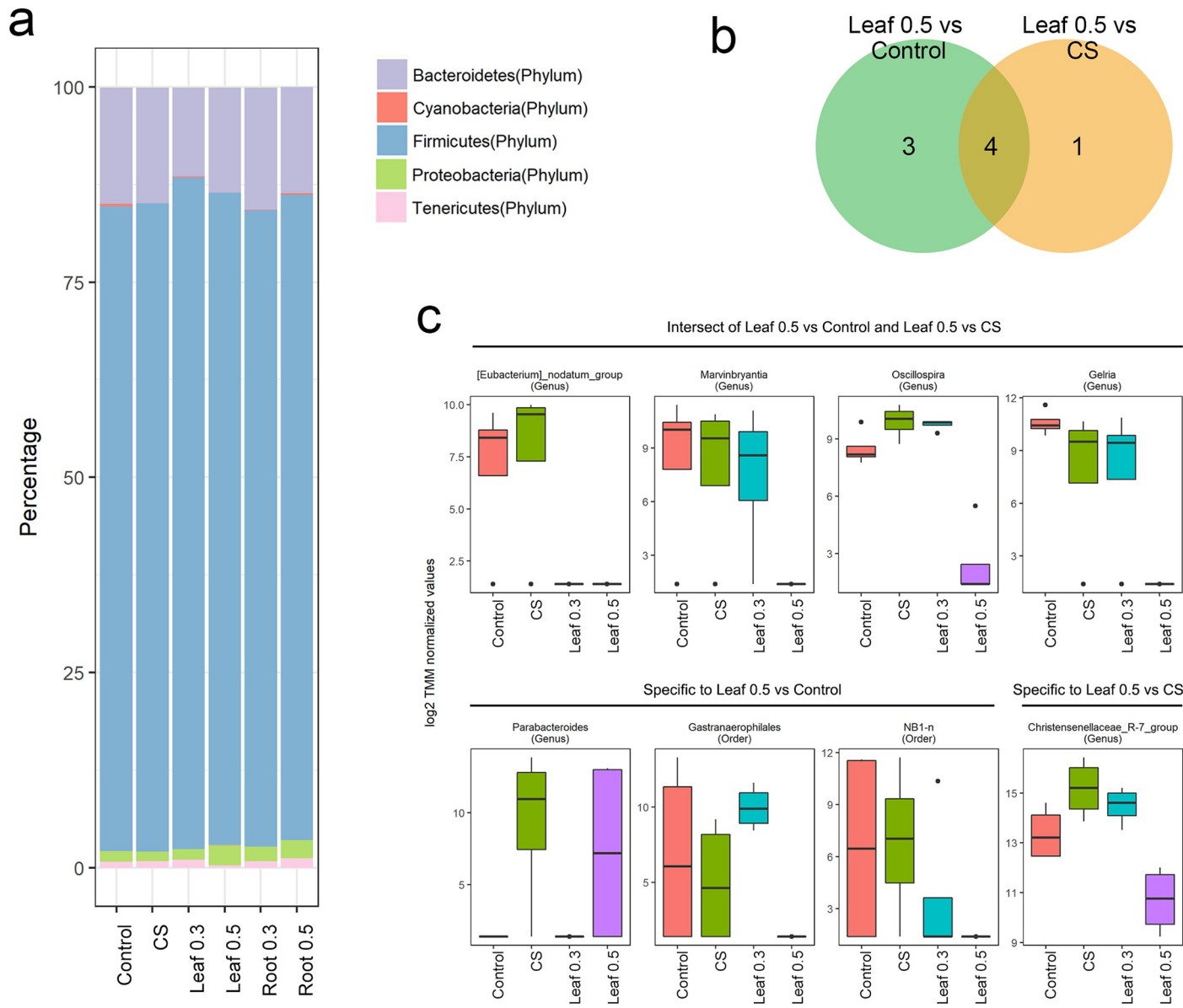

**Fig 2. Abundance of microorganisms at phylum and genus levels.** (a) Phylum level composition: Bar plots represent the percentage (%) of average abundance for groups. (b) Venn diagrams showing differential abundance of features at the genus level. (c) The Leaf group shows the relative abundance of different taxa.

The abundance of a few common genera varied in Control vs. Leaf 0.5, CS vs. Leaf 0.5, and CS vs Leaf 0.3. *Parabacteriodes* and *Eubacterium nodatum* group, which differed between chickens exposed to CS and Leaf 0.3 also varied in abundance in the Control and Leaf 0.5 groups. Most genera included in Leaf 0.5 compared with CS were also identified in Leaf 0.5 compared with Control. *Eubacterium nodatum group*, *Marvinbryantia*, *Oscillospira*, and *Gelria* species were common in both tests (Fig 2B). Significant and differentially abundant genera detected only in Leaf 0.5 compared with Control included *Parabacteroides*, *Gastranaerophilaes*, and *NB1-n* while the *Christensenellacea R-7 group* was found abundantly only in Leaf 0.5 compared with CS. We observed a decrease in the abundance of most of the genera except for *Parabacteroides* (Fig 2C). *Eubacterium nodatum group* showed almost undetected in chickens

fed with Leaf 0.3 and Leaf 0.5. The abundance of *Marvinbryantia*, *Gelria* and *NB1-n* decreased gradually according to the amount of Leaf supplementation compared with their mean abundance in the Control. A rapid reduction in the levels of *Oscillospira*, *Gastranaerophilales* and *Christensenellaceae R-7 group* was observed in chickens exposed to Leaf 0.5.

## Correlation between the abundance of microorganisms and growth traits

*A. hookeri* not only affects the microbiome composition, but also the growth traits such as body weight, bone strength, and levels of infectious bursal disease (IBD) antibody. These traits were linked to gut microbiome in several studies [27,28]. The abundance of genera was correlated with growth traits. Spearman's rank correlation coefficients were significant for 7, 5, and 7 genera with body weight, bone strength, and IBD antibody, respectively (Spearman's rho; P-value < 0.05).

Among the seven genera correlating with body weight, *Ruminococcaceae UCG-005* showed highly negative correlation with body weight (Spearman's rho = -0.56) (Fig 3A). *Clostridium sensustricto 1* had the highest correlation with body weight (Spearman's rho = 0.44). The relative abundance of correlated genera was indicated next to the correlation index. The genera showing negative correlation with body weight had lower abundance in Leaf 0.3 and Leaf 0.5 categories while the genera displaying positive correlation with body weight showed relatively higher abundance in Leaf 0.3 and Leaf 0.5 groups. Thus, the relative abundance of microorganisms correlated with body weight is related to the specific Leaf diet.

Most of the genera that correlated with bone strength were negative (Fig 3B). Most microorganisms that negatively correlated with bone strength showed a lower abundance in chickens fed with Leaf 0.5 compared with those in other groups. *NB1-n* genera showing negative correlation with bone strength (Spearman's rho = -0.43) were significantly lower in Leaf 0.5. However, *Parasutterella* only showed a highly positive correlation with bone strength (Spearman's rho = 0.74).

Seven microorganisms correlated with antibody titers against IBD, including three species that were negatively correlated and four that were positively correlated (Fig 3C). The *Eubacterium brachy group* showed the highest correlation (Spearman's rho = 0.57), showing increased abundance in Leaf 0.3 and Leaf 0.5 but uniformly lower abundance in other groups. *Parasutterella* was positively correlated with bone strength and with IBD antibody production. The abundance of *Gelria* was significantly decreased in chickens fed with Leaf 0.5 and showed a negative correlation with IBD antibody (Spearman's rho = -0.44). Exposure to Leaf 0.5 also resulted in a similar pattern of microbial abundance correlated with IBD antibody, with a higher (or lower) abundance in chickens fed with Leaf 0.5 showing positive (or negative) correlation with IBD antibody.

## Prediction of gut microflora function

Microbial functions predicted by phylogenetic investigation of communities via reconstruction of unobserved states (PICRUSt) were compared between chickens exposed to Control and Leaf 0.5 because the genera correlating with growth traits were associated with their abundance in Leaf 0.5 (Fig 4). Among the 38 different functions analyzed, 30 were associated with a lower abundance and eight with a higher abundance in chickens fed with Leaf 0.5. Most higher functions in Leaf 0.5 were related to metabolism. Enriched functions of microorganisms in Leaf 0.5 group such as C5-branched dibasic acid metabolism, fructose and mannose metabolism, and galactose metabolism were included in carbohydrate metabolism. Most significantly, lysine degradation was depleted in Leaf 0.5. Among the functions depleted in chickens exposed to Leaf 0.5, the terms related to human disease were prevalent (S3 Fig). Especially, the

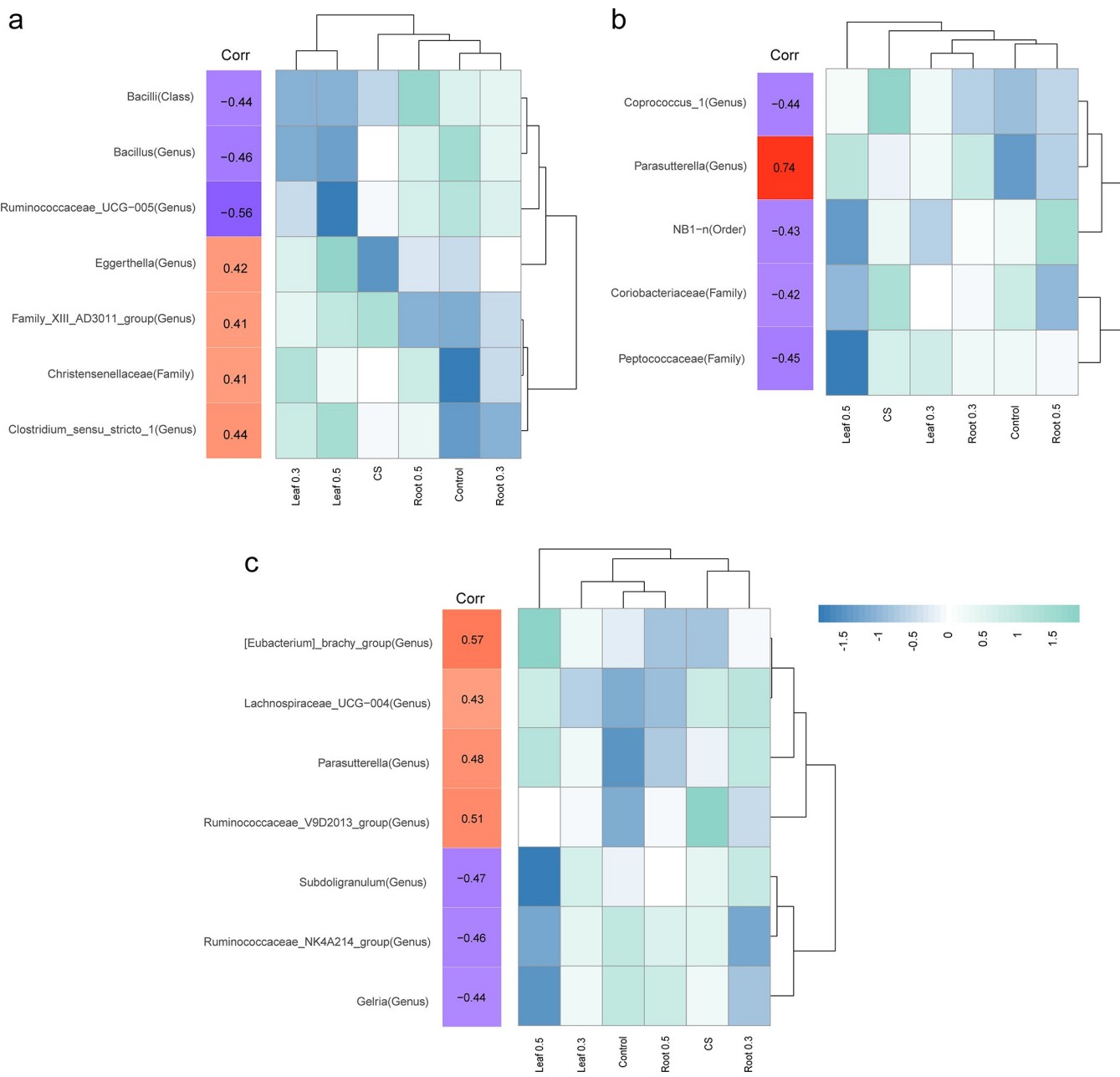

**Fig 3. Correlation between microbiota and growth performance including body weight, bone strength, and IBD antibody.** (a) Microorganisms showing significant correlation with body weight: The column in "corr" shows Pearson's. r value. The heatmap represents the mean abundance of groups. (b) Microorganisms showed significant correlation with bone strength. (c) Microorganisms showed significant correlation with antibody titers against infectious bursal disease (IBD).

functions of microorganisms depleted in Leaf 0.5 category related to pathways in cancer, renal cell carcinoma, colorectal cancer, and small cell lung cancer.

## Performance analysis

In the present study, body weight, bone strength, and IBD were measured. The Leaf 0.3 and Leaf 0.5 groups showed higher body weights (P-value: 0.001 and 0.052, respectively) compared

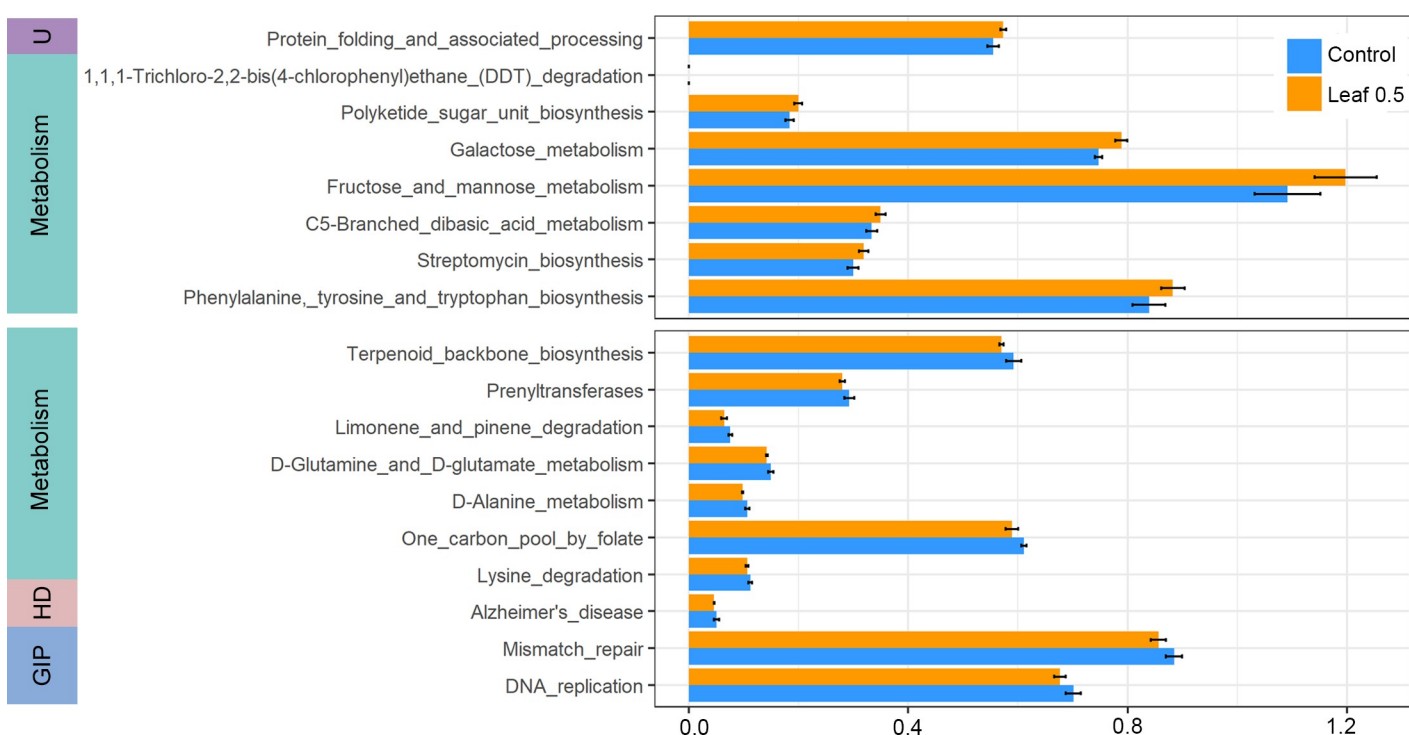

**Fig 4. Predicted functions of microorganisms showing significant difference between Control and Leaf 0.5.** Functions were predicted using PICRUSt. Functions with significant difference (P < 0.05, one sided Wilcoxon rank sum test) are shown. The error bar represents SD (standard deviation). The upper box shows functions enriched in Leaf 0.5 compared with control while the lower box shows functions enriched in control compared with Leaf 0.5 with top 10 p-value. U: Unclassified, HD: Human Diseases, GIP: Genetic Information Processing.

with the control group (S4 Fig). Bone strength and IBD showed higher levels in the Leaf 0.3 and Leaf 0.5 groups.

## Discussion

This study demonstrated the effect of *A. hookeri* on gut microbiome, especially in the cecum of broiler chicken. In this study, we fed different feed supplements including *A. hookeri* to six groups of 1,200 broiler chickens, each group comprising 200 chickens: Control, CS, Leaf 0.3, Leaf 0.5, Root 0.3 and Root 0.5. Four chickens from each of the six groups with their weights similar to the average weight of the group were selected. Twenty-four samples from six groups were used for sequencing. The impact of Leaf 0.5 on the altered gut microbiome in this study may corroborate our previous study, which highlighted improved growth traits in Leaf 0.5. Based on multiple analysis of gut microbiome and phenotype of broiler chickens, this study provides deeper insight into the relationship between the intake of *A. hookeri* Leaf, microbial communities and growth performance.

The remarkable effect of leaf on microbiome is attributed to the abundance of pharmacologically active components contained in leaves. Previous studies regarding *A. hookeri* compared the concentrations of different ingredients in leaf with those of the root. Compared with the root, the leaf contains a higher number of polyphenols, estimated by the amount of total phenolics and flavonoids [15]. Polyphenol is an antioxidant that potentially prevents diseases related to oxidative stress, including cancer and cardiovascular diseases [29]. A large proportion (~90%) of absorbable polyphenols are digested by gut microbiome rather than the digestive enzymes in the small intestine [30]. Microbial degradation of polyphenols generates

intermediates including aglycones that promote various aromatic acid metabolites [31]. The altered metabolite composition may alter the prebiotic effects of microrganisms such as anti-microbial activities against pathogens [32]. Especially, the reduction of bacteria was observed only in chickens exposed to leaves, which showed higher levels of total phenolics (240.4–276.6 mg gallic acid equivalents/100 g) compared with the roots [15]. In broiler chickens, polyphenol-rich grapes showed increased *Enterococcus* levels but decreased levels of *Clostridium* [21]. The polyphenols in *A. hookeri* might play an important role in modulating microorganism composition, which explains the presence of unique microbial communities in the guts of chickens fed with polyphenol-rich leaves.

The altered abundance of bacteria following *A. hookeri* leaf supplementation occurs at the generic level. Among eight leaf-associated genera, the abundance of *Parabacteroides* was increased in the Leaf 0.5 group. *Parabacteroides* are obligatory anaerobic bacteria that degrade saccharides, forming acetate and succinate as the major end-products [33]. Except for *Parabacteroides*, the abundance of *A. hookeri*-associated genera decreased. The reduction in the abundance of most microbiomes may be associated with the antibacterial effect of *A. hookeri* leaf. The decrease in bacterial concentrations may be attributed to allicin, which accounts for the spicy flavor of *A. hookeri*. Allicin is an organosulfur compound with characteristic antibacterial activity against a wide range of microorganisms including *Staphylococcus* and *Pseudomonas* in several studies [23,34]. Its antibacterial activity is attributed to its chemical reaction with thiol groups of enzymes that influence the metabolism of cysteine proteinase activity related to bacterial virulence [35]. However, the mechanisms of antibacterial activity of *A. hookeri* are not clear. Therefore, further evidence is needed to establish the relationship between the decreased abundance of microorganisms and the antibacterial activity of *A. hookeri*.

The effects of *A. hookeri* on crucial phenotypes of broiler chickens were investigated, given the importance of growth trait enhancement without antibiotics for growth promotion (AGPs) and to decrease the risk of antibiotic resistance [36]. Other studies investigated feed supplements using natural foods including extracts from herb species, which enhanced poultry health by improving immunity and protecting chickens from avian diseases [37–44]. Our previous study has shown that broiler chickens supplemented with *A. hookeri* show improved growth performance [22]. In the present study, a subset of samples from our previous study was used [45]. The Leaf 0.3 and Leaf 0.5 groups showed higher body weights (P-value: 0.001 and 0.052, respectively) compared with the control group (S4 Fig). To determine whether the microbiome was related to greater body weight in chickens exposed to *A. hookeri* leaves and to identify the genera associated with body weights, the abundance of genera was correlated with body weight. Notably, the genera showing positive (or negative) correlation with body weight were relatively abundant (or depleted) in Leaf 0.3 and Leaf 0.5 groups, suggesting that alterations in microbiome induced by leaf exposure may affect the body weight. Specific genera related to body weight are known to be related to diet or energy metabolism in other studies. For example, *Bacillus* that showed negative correlation with body weight was associated with feed efficiency in broiler chicks [46]. *Clostridium sensustricto* 1 showed a positive correlation with body weight. Family *Clostridiaceae* is known to induce weight gain in rex rabbits [47]. The association between increased body weight and microbiome is also supported by functional analysis. Results of functional analysis revealed that the chickens in Leaf 0.5 group showed enriched carbohydrate metabolism, which increased the body weight. In summary, the correlation between body weight and enrichment of carbohydrate metabolism suggests that altered communities of microorganisms induced by *A. hookeri* may alter the body weight.

The association between microbiome and other traits including bone strength and IBD antibody was also determined. Bone strength was examined because several bacteria improved calcium absorption [48]. IBD antibody was investigated as an index of immune system related

to microbiome [24]. Among five genera associated with bone strength, none of them was correlated with bone strength in previous studies. However, the genera associated with IBD antibody were consistent with or contrary to previous metagenome studies related to immune system or IBD. *Lachnoclostridium* and *Ruminococcaceae* groups that were positively and negatively correlated with IBD antibody, respectively, were also involved in the inoculation of strongly virulent IBD virus in broiler chickens [49]. This finding indicates that these genera are strongly linked to the regulation of immune system.

Parasutterella, which showed a positive correlation with bone strength was also positively correlated with IBD. *Parasutterella* showed positive correlation with irritable bowel syndrome in a previous study [8]. However, the role of *Parasutterella* in gut pathophysiology has yet to be discovered. We speculated that the association with bone strength and IBD in common genera might be explained by the relationship between immune and bone cells. The interaction is mediated via circulating blood cells containing T and B cells in bone marrow [28]. T and B cells play important roles in cell-mediated immunity to regulate bone resorption and bone formation by producing large amounts of cytokines [50], which explains the connection between bone and immune cells. A mouse study has revealed that gut microbiota regulate bone mass and immune status [51]. Despite the association between specific genera and both traits, microorganisms known to control both bone mass and immune system were poorly detected. Thus, the genera detected in this study may provide a clue for this interaction. The small sample size is one of the study limitations; however, this is the first study to investigate the effect of *A. hookeri* on gut microbiome, despite several studies reporting its health benefits. Further studies are needed to determine the role of these genera in the interaction with immune system and bone strength.

In conclusion, our results showed the degree of alteration based on supplementation of *A. hookeri*, especially its leaf. The abundance of microorganisms was decreased in chickens exposed to *A. hookeri* leaves indicating that the leaf has a distinct effect on microbial communities. We also detected specific genera related to body weight, bone strength, and IBD antibody known to be important for productivity of broiler chickens. Therefore, the health benefit of *A. hookeri* in broiler chicken is mediated via its microbiome, suggesting that *A. hookeri* is a potential feed supplement for broiler chickens.

## Materials and methods

### Sample collection

Experimental protocol and procedures were approved by the Small Animal Care and Use Committee of the National Institute of Agricultural Sciences (NAS201707). A total of 1200 male broiler chickens (Arbor Acres broilers) were grown for 35 days. They were divided into six groups (n = 200 chickens/group): Control, CS, Leaf 0.3, Leaf 0.5, Root 0.3, and Root 0.5. All groups were freely fed with a basal diet (crude protein (CP) 22%, metabolic energy (ME) 3,100 kcal/kg for 0–3 weeks; and CP 20%, ME 3,150 kcal/kg for 3–5 weeks). A diet containing 0.05% Commercial Xtract (ML Co, Seoul, Korea) was used in the CS group. *A. hookeri* leaf or root powder (0.3% or 0.5%) was added to Leaf 0.3, Leaf 0.5, Root 0.3, and Root 0.5 groups. The different powders of *A.hookeri* were prepared after freeze-drying and grinding. Four chickens in each of the six groups similar to the average group weight were selected. Thus, a total of 24 samples (four samples from each of the six groups) were used for sequencing.

### DNA extraction and Illumina Sequencing

Cecal samples were used for DNA extraction using AccuPrep Stool DNA Extraction Kit following the manufacturer's instructions. The V3 and V4 regions of the 16S rRNA genes were

PCR-amplified from the microbial genomic DNA. The DNA quality was determined using PicoGreen and Nonodrop methods. The input gDNA (10 ng) was PCR-amplified using the barcoded fusion primers 341F/805R (341F: 5′ CCTACGGGNGGCWGCAG 3, 805R: 5′ GA CTACHVGGGTATCTAATCC 3′). The final purified product was quantified using qPCR according to the qPCR Quantification Protocol Guide (KAPA Library Quantification kits for Illumina Sequencing platforms) and qualified using the LabChip GX HT DNA High Sensitivity Kit (PerkinElmer, Massachusetts, USA). The 300 paired-end sequencing reaction was performed on MiSeq™ platform (Illumina, San Diego, USA). The sequencing data were deposited into the Sequence Read Archive (SRA) of NCBI (http://www.ncbi.nlm.nih.gov/sra) and accessed via accession number SRP151247.

## Taxonomic analysis

De-multiplexed paired-end reads were merged with PEAR [52]. Pre-processed reads were analyzed using QIIME2 version 2017.12. We used DADA2 software package [53] implemented in QIIME2 to model and correct Illumina-sequenced FASTAQ files by removing chimeras using "consensus" method. QIIME2 q2-feature-classifier plugin was trained on a Silva database (Release 128) for 99% OTU full-length sequences.

Alpha and beta-diversity analyses were performed with q2-diversity plugin in QIIME2 at a sampling depth of 1000. Weighted Unifrac distance matrix was used for permutation multivariate analysis of variance (PERMANOVA) and PCoA plots. PERMANOVA was performed with 999 permutations to weighted UniFrac distance matrix using Adonis in R package 'vegan' [54].

## Identification of differentially abundant microbiomes (DAM)

Trimmed Mean of M values (TMM) were obtained to adjust for different library sizes using edgeR [55]. Statistical tests were performed under generalized linear model (GLM) considering OTU counts as negative binomial distribution. To compare the goodness-of-fit of two models, the log-likelihood ratio was calculated. In the statistical test, the false discovery rate (FDR) was used to adjust for multiple testing errors with a significance level of 5% [56].

## Performance analysis

In the present study, body weight, bone strength, and IBD antibody level of experimental chicken were measured at 5 weeks of age [45]. Four chickens in each of the six groups similar to the average group weight were selected for the Performance analysis. Thus, a total of 24 samples (four samples from each of the six groups) were used for bone strength and IBD antibody level of blood collected from heart. Tibia strength was measured using instron (Model 3342, Inatron Universal Testing Machine, Instron Corp., Norwood, MA, USA). Effects of *A. hookeri* on antibody titer against IBD in growing chickens were evaluated by Hemmagglutination inhibition test (HI test).

## Statistical analysis of growth traits

The Student's t-test was used to compare the growth traits between groups. The correlation between microbiota and growth traits was determined via Spearman correlation between TMM values of each sample and growth traits (body weight, bone strength and IBD antibody). The abundance (TMM value) of significantly correlated genera was visualized in a heat-map using pheatmap R package.

### PICRUSt analysis and statistical comparison of functions between groups

Phylogenetic investigation of communities by reconstruction of unobserved states (PICRUSt) was used to predict functional profile of microbiota [57]. Since PICRUSt uses a closed reference OTU picking based on Greengenes database (version 13.5.), the features assigned to Greengenes databases were used. The abundance of functions in Control was compared to that in Leaf 0.5 using Wilcoxon rank-sum test.

## Supporting information

**S1 Table. Results of PERMANOVA of pair-wise test for combinations groups.**
(PDF)

**S1 Fig. Shannon index and observed OTUs in six groups.** In Shannon index, a significant difference was observed between Leaf 0.3 and Root 0.3 group compared with CS ($P = 0.02$). In the observed OTU, Leaf 0.3 differed significantly from CS ($P = 0.02$).
(TIF)

**S2 Fig. Phylum level composition.** Bar plots represent the percentage (%) of average abundance among groups.
(JPG)

**S3 Fig. Predicted microbial functions deficient in Leaf 0.5.** Functions were predicted by PICRUSt. All functions enriched in Control compared with Leaf 0.5 are presented (P-value < 0.05). Terms are separated into upper and lower boxes because of scale.
(TIF)

**S4 Fig. Body weight, bone strength, and IBD antibody of broiler chickens.** Leaf 0.3 and Leaf 0.5 showed higher body weight (P-value: 0.001 and 0.052, respectively) compared with the control group.
(TIF)

## Acknowledgments

The authors appreciate Sung-Moon Jo for assistance with the care of chickens and C&K genomics for helping sample analysis, and thank Dr. You-Suk Kim for his critical review.

## Author Contributions

**Conceptualization:** Sung-Hyen Lee.

**Data curation:** Sung-Hyen Lee.

**Formal analysis:** Sung-Hyen Lee, Sohyun Bang, Bong-Sang Kim, Seung-Hwan Kim, Sang-Hyun Kang, Kyung-Woo Lee, Dong-Wook Kim.

**Funding acquisition:** Sung-Hyen Lee, Hwan-Hee Jang, Jeong-Sook Choe.

**Investigation:** Sung-Hyen Lee, Seung-Hwan Kim.

**Methodology:** Sung-Hyen Lee, Sohyun Bang, Bong-Sang Kim, Seung-Hwan Kim, Kyung-Woo Lee, Dong-Wook Kim.

**Project administration:** Sung-Hyen Lee.

**Resources:** Sung-Hyen Lee, Sang-Hyun Kang.

**Software:** Sung-Hyen Lee.

**Supervision:** Sung-Hyen Lee, Bong-Sang Kim, Jung-Bong Kim.

**Validation:** Sung-Hyen Lee, Sohyun Bang.

**Visualization:** Sung-Hyen Lee.

**Writing – original draft:** Sung-Hyen Lee, Sohyun Bang.

**Writing – review & editing:** Sung-Hyen Lee, Sohyun Bang, Hwan-Hee Jang, Eun-Byeol Lee, Jeong-Sook Choe, Shin-Young Park, Hyun S. Lillehoj.

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
