## [Decision Letter · Decision Letter 0]

15 Aug 2019

PONE-D-19-17223

Effects of Allium hookeri leaf on gut microbiome related to growth performance of young broiler chickens

PLOS ONE

Dear Dr Lee,

Thank you for submitting your manuscript to PLOS ONE. After careful consideration, we feel that it has merit but does not fully meet PLOS ONE’s publication criteria as it currently stands. Therefore, we invite you to submit a revised version of the manuscript that addresses the points raised during the review process.

We would appreciate receiving your revised manuscript by Sep 29 2019 11:59PM. To enhance the reproducibility of your results, we recommend that if applicable you deposit your laboratory protocols in protocols.io, where a protocol can be assigned its own identifier (DOI) such that it can be cited independently in the future. For instructions see: http://journals.plos.org/plosone/s/submission-guidelines#loc-laboratory-protocols

We look forward to receiving your revised manuscript.

Kind regards,

Juan J Loor

Academic Editor

PLOS ONE

Journal Requirements:

This work was supported by Agriculture Science & Technology Development (PJ01178705, PJ01327901) funded by Rural Development Administration, Republic of Korea. The fund was used in study design, data collection and analysis, or preparation of the manuscript.

We note that one or more of the authors are employed by a commercial company: C&K genomics and BHNBIO.

Reviewers' comments:

Reviewer's Responses to Questions

**Comments to the Author**

1. Is the manuscript technically sound, and do the data support the conclusions?

Reviewer #1: Partly

2. Has the statistical analysis been performed appropriately and rigorously? 

Reviewer #1: Yes

3. Have the authors made all data underlying the findings in their manuscript fully available?

Reviewer #1: Yes

4. Is the manuscript presented in an intelligible fashion and written in standard English?

Reviewer #1: Yes

5. Review Comments to the Author

Reviewer #1: The article has scientific merit. However, it needs some adjustments and some clarification.

The title should contain the word "root", not just leaf

Important information is missing from the summary as the main bacteria found, what were the main changes with the treatments. Still in the summary, the authors report that the main changes are involved with the 0.5% level, but I disagree, throughout the work it is possible to realize that 0.3% has a great impact on the results.In the conclusion of the summary it must be clear that the effect was of the leaf.

At the end of the introduction the goal needs to be clearer, it went far beyond the profile of the microbiota. Several correlations were presented that were also objective.

In line 128, the authors use the term microbes, the advised term is the microorganisms please note this throughout the text.

Data on the composition (profile) at the genus level should be showed for all treatments this will make it easier to understand figures 2b and 2c.

In the discussion, between lines 232 and 238 the authors report that A.hookeri has effect as a prebiotic, however, they do not make it clear.

But if it has prebiotic action what would be the leaf component responsible for this effect?And how much of this compound is in the leaf?

In line 251 boiler = broiler

In line 256 the authors report the following sentence "In the present study a subset of samples from our previous study was used" two points here are crucial. First: the methodology for performance analysis should be in the material and methods as well as bone and IBD results should be presented. What statistical analyzes were used for these data? Second: If these performance data were performed in a separate experiment; Correlation between the data is not feasible because we know that the microbiota is variable.

The discussion about the microbiota and its effect on performance, ibd and bones was short. We know that when the immune system is stimulated the impact on performance is negative. If Parasutterella stimulates the immune system, how can it be treated for performance?

6. PLOS authors have the option to publish the peer review history of their article (what does this mean?). If published, this will include your full peer review and any attached files.

Reviewer #1: Yes: Priscila de Oliveira Moraes

---

## [Author Response · Author response to Decision Letter 0]

1 Nov 2019

Attached response letter in the "Attach files section".

---

## [Decision Letter · Decision Letter 1]

9 Dec 2019

Effects of Allium hookeri on gut microbiome related to growth performance in young broiler chickens

PONE-D-19-17223R1

Dear Dr. Lee,

We are pleased to inform you that your manuscript has been judged scientifically suitable for publication and will be formally accepted for publication once it complies with all outstanding technical requirements.

With kind regards,

Juan J Loor

Academic Editor

PLOS ONE

Additional Editor Comments (optional):

Reviewers' comments:

Reviewer's Responses to Questions

**Comments to the Author**

1. If the authors have adequately addressed your comments raised in a previous round of review and you feel that this manuscript is now acceptable for publication, you may indicate that here to bypass the “Comments to the Author” section, enter your conflict of interest statement in the “Confidential to Editor” section, and submit your "Accept" recommendation.

Reviewer #1: All comments have been addressed

2. Is the manuscript technically sound, and do the data support the conclusions?

Reviewer #1: Yes

3. Has the statistical analysis been performed appropriately and rigorously? 

Reviewer #1: Yes

4. Have the authors made all data underlying the findings in their manuscript fully available?

Reviewer #1: Yes

5. Is the manuscript presented in an intelligible fashion and written in standard English?

Reviewer #1: Yes

6. Review Comments to the Author

Reviewer #1: (No Response)

7. PLOS authors have the option to publish the peer review history of their article (what does this mean?). If published, this will include your full peer review and any attached files.

Reviewer #1: No

---

## [Editor Report · Acceptance letter]

23 Dec 2019

PONE-D-19-17223R1 

Effects of *Allium hookeri* on gut microbiome related to growth performance in young broiler chickens 

Dear Dr. Lee:

I am pleased to inform you that your manuscript has been deemed suitable for publication in PLOS ONE. Congratulations! Your manuscript is now with our production department. 

With kind regards,

on behalf of

Dr. Juan J Loor 

Academic Editor

PLOS ONE